# Covalent Organic Frameworks: Synthesis, Properties and Applications—An Overview

**DOI:** 10.3390/polym13060970

**Published:** 2021-03-22

**Authors:** Tiago F. Machado, M. Elisa Silva Serra, Dina Murtinho, Artur J. M. Valente, Mu. Naushad

**Affiliations:** 1University of Coimbra, CQC, Department of Chemistry, 3004-535 Coimbra, Portugal; tiago.f.machado@hotmail.com (T.F.M.); melisa@ci.uc.pt (M.E.S.S.); dmurtinho@ci.uc.pt (D.M.); 2Advanced Materials Research Chair, Department of Chemistry, College of Science, King Saud University, Riyadh 11451, Saudi Arabia; mnaushad@ksu.edu.sa; 3Yonsei Frontier Lab, Yonsei University, Seoul 03722, Korea

**Keywords:** COFs, Covalent Organic Frameworks, microporous polymers

## Abstract

Covalent Organic Frameworks (COFs) are an exciting new class of microporous polymers with unprecedented properties in organic material chemistry. They are generally built from rigid, geometrically defined organic building blocks resulting in robust, covalently bonded crystalline networks that extend in two or three dimensions. By strategically combining monomers with specific structures and properties, synthesized COF materials can be fine-tuned and controlled at the atomic level, with unparalleled precision on intrapore chemical environment; moreover, the unusually high pore accessibility allows for easy post-synthetic pore wall modification after the COF is synthesized. Overall, COFs combine high, permanent porosity and surface area with high thermal and chemical stability, crystallinity and customizability, making them ideal candidates for a myriad of promising new solutions in a vast number of scientific fields, with widely varying applications such as gas adsorption and storage, pollutant removal, degradation and separation, advanced filtration, heterogeneous catalysis, chemical sensing, biomedical applications, energy storage and production and a vast array of optoelectronic solutions. This review attempts to give a brief insight on COF history, the overall strategies and techniques for rational COF synthesis and post-synthetic functionalization, as well as a glance at the exponentially growing field of COF research, summarizing their main properties and introducing the numerous technological and industrial state of the art applications, with noteworthy examples found in the literature.

## 1. Reticular Chemistry and MOFs

Reticular chemistry is a recently developed area in organic chemistry and material science, concerning the strategic and rational synthesis of new organic compounds by promoting the controlled growth and formation of regular, porous and crystalline structures [1,2]. By choosing adequate building blocks with well-defined symmetry, it is possible to create solids with a regular, predetermined structure, based on repeating subunits with specific geometrical properties (secondary building units (SBUs)) that may be interesting from an industrial or technological standpoint [3].

The first major result of the application of reticular chemistry concepts was the development of a new class of materials, so-called Metal Organic Frameworks (MOFs). These new structures, branching out from the family of coordination polymers, consist in the coordination between metal ions (or clusters) and multifunctional organic ligands, capable of forming subunits that extend in two or three dimensions [1].

In a MOF, the metal ions act like coordination points, or vertices, while the organic ligands, whose structures are typically rigid and spatially well-defined, are the backbone of the reticulated 3D structure.

The first synthesized MOF structure (MOF-5) was prepared by Li and coworkers from the coordination of Zn ions with dicarboxylate ligands [4]. In this framework, the connecting points consist of Zn_4_(O)O_12_C_6_ clusters in which an oxygen atom is surrounded by four Zn^2+^ atoms, which in turn are coordinated to six carboxylic groups from the six 1,4-benzenedicarboxylate (DBC) ligands. Since the ligands are symmetrical and bifunctional, each one is further coordinating with two other adjacent zinc ions from nearby clusters, resulting in a regular, permanent cubic-shaped structure [4].

MOFs are described as crystalline, rigid, highly stable and highly porous materials [5]. Depending on the ligands and corresponding tridimensional structures, it is possible to control the crystalline framework and the shape of the pores for a wide range of specific, highly directed applications, such as adsorption of gas and pollutants, energy storage, chemical sensing or heterogeneous catalysis [4,6,7,8].

## 2. COFs—Covalent Organic Frameworks

The emergence and development of reticular chemistry in organic synthesis gave birth to another class of materials, called Covalent Organic Frameworks (COFs). Unlike MOFs, which are characteristically hybrid by their nature, integrating both organic and inorganic elements in their structures, COFs are entirely organic, consisting mainly of light elements from the periodic table (H, B, C, N and O). Additionally, while MOFs are built on the coordination between metals and ligands, COF building blocks are covalently bonded, creating a comparatively stronger and more robust 3D crystalline framework [1,2,9,10].

COFs have recently been gaining the curiosity of the scientific community because of their unusual properties and the potential they have been showing for the synthesis of highly specific, targeted materials with spatial precision at the atomic scale [11]. The directionality and bond strength common to covalent bonds allow for the formation of unprecedented organic structures in the universe of polymer chemistry, resulting in materials with rigid, low density, as well as being highly stable and highly resistant, with well-defined, permanent porosity. In fact, even though they are a very recent new class of materials, COFs already present results which are capable of rivaling, if not even surpassing, the results obtained for more developed analogous classes of reticulated structures, namely zeolites (a class of inorganic structures), MOFs (metal-organic hybrid class), or several other (typically less crystalline) organic polymer subclasses that have been developed, such as PIMs (Polymers with Intrinsic Microporosity), HCPs (Hyper crosslinked Polymers), CMPs (Conjugated Microporous Polymers), and PAFs (Porous Aromatic Frameworks), among others [9,11].

The structure of the resulting crystalline framework will depend on the geometry of the chosen building blocks, which consist of multifunctional monomers, with defined structure and symmetry, creating regular structures that extend in two and three dimensions. The condensation of polymers of different geometries and point group symmetries will produce structures of varying topologies, with different degrees of reversibility, which in turn will result in a wide range of pore sizes, shapes and distributions throughout the material [10,12].

The reticular condensation between, for example, a trifunctional C_3_ monomer with a bifunctional C_2_ monomer will result in a theoretical hexagonal framework, of *hcb* (*honeycomb*) topology. If the C_3_ monomer is replaced by an analogous C_4_ structure, the resulting material will show square-shaped pores and *sql* (*square lattice*) topology [10,12].

COFs are labeled 2D or 3D according to the dimensional structure of the theoretical unit cell used to represent the crystalline framework. PI-COF, an imide-type structure built from tris(4-aminophenyl)amine (TPA) and pyromellitic dianhydride (PMDA) is an example of a 2D COF [13] (Figure 1). The 3D structure of the real synthesized material consists of the close packing of those planes, through dipolar intermolecular interactions or, in the specific case of aromatic-rich structures, π orbital stacking.

If one of the monomers is 3D, the resulting COF will be 3D. COF-320, an imine-type structure synthesized from tetrakis(4-aminophenyl)methane (TAPM) and 4,4’-biphenyldicarbaldehyde (BPDA) is an example of a 3D COF [14] (Figure 1).

The first COFs reported in the literature date back to 2005 and were synthesized by Côté et al. by reacting boronic acid derivatives [15]. COF-1, one of the synthesized materials, was obtained through the self-condensation of diboronic acid (BDBA), yielding a material described by a 2D network of hexagonal pores made from boroxine-type bonds (B_3_O_3_ rings) of approximately 1.5 nm in diameter [15]. The other compound, COF-5, was prepared by reacting BDBA and 2,3,6,7,10,11-hexahydroxytriphenylene (HHTP), forming a reticulated structure of boronic ester linkages (Scheme 1). The rotating axes of symmetry of the planar monomers (C_2_ and C_3_, respectively) allow for a framework similar to COF-1, with equally hexagonal, permanent pores, but with larger diameter (2.7 nm), which is a direct consequence of the larger dimensions of the building blocks employed in the synthesis. Both COF-1 and COF-5 showed relatively low density, high surface area and a thermal stability of up to 600 °C, which is highly unusual for entirely organic structures [15].

In practice, however, the formation of an amorphous material should be the most thermodynamically probable outcome. Inducing order to the structure throughout the synthetic process is, therefore, one of the main challenges in efficient COF synthesis [9,11,15]. A possible approach to attain this consists in controlling the reversibility of the condensation reaction [16]. In the case of COF-5, retaining water within the reactant mixture is fundamental, since it lets the reaction seamlessly revert back to the starting monomers, which allows for the “correction” of conformational “errors” within the polymer structure, thus enabling a more ordered framework to emerge. Obtaining a crystalline final product depends, then, on establishing an efficient equilibrium between the kinetic and thermodynamic synthetic processes [16]. This implies, in general, high temperatures, an inert, sealed environment and long reaction times [1,11].

## 3. COF Linkage Types—Structure and Properties

A wide range of monomers have been investigated to prepare new COFs with varying properties and applications, based on different types of chemical bonds and linkages. Among those, COFs bearing imine, hydrazone, azine, imide and β-ketoenamine are particularly interesting [11].

Imine-type COFs are obtained by condensing polyaldehydes with polyamines. They are more stable than boronic ester bondbearing COFs, since they are, in general, more resistant to hydrolysis [11].

The first imine-type COF, termed COF-300, was obtained by Uribe-Romo and coworkers by condensation between TAPM and terephthalaldehyde (BDA) [17] (Figure 2). The structure also shows several other intertwining crystalline forms, due to the long bond length between the two *sp^3^* carbon atoms from the tetramine. Similarly to other COFs, COF-300 is insoluble in water and common organic solvents and highly porous, with pores 0.72 nm in diameter and thermal stability of up to 490 °C [17].

COF-LZU1 was the first imine-type COF bearing a 2D unit cell and was synthesized by Ding et al. from 1,3,5-triformylbenzene (TFB) and 4-phenylenediamine (PDA) [18]. The material has 1.8 nm diameter pores and, in its eclipsed structure, is especially nitrogen-rich, with N atoms from adjacent layers being only 3.7 Å apart, which allows for favorable metal ion coordination. This property was taken advantage of and thus COF-LZU1 became the first COF structure to be tested in heterogeneous catalysis, with excellent conversion rates [18].

Concerning the polymerization mechanism, it has recently been reported that, unlike boronic ester-type COFs, which gradually grow from the first established crystal cores, COFs bearing imine groups grow through different, independent, isolable fragments which then merge to create supramolecular structures [19].

Much like imine-type COFs, COFs with hydrazone and azine linkages are synthesized by the reversible reaction between symmetric aldehydes and hydrazides or hydrazine, respectively. Uribe-Romo and coworkers reported the synthesis of the first two COFs with hydrazone functionality, by reacting 2,5-diethoxyterephthalohydrazide (DETH) and trigonal trialdehydes of different dimensions [20]. One of the materials, COF-42, obtained from DETH and TFB (Figure 3), shows thermal stability of up to 280 °C. Since the hydrazone bond reversibility is particularly pH dependent, the crystallization process can be controlled through dynamic synthetic conditions [1]. Due to the bigger size of the building monomers, COF-42 showed 2.8 nm pores and a Brunauer-Emmett-Teller (BET) surface area of 720 m^2^ g^−1^ [20].

Azine-type COFs are formed by condensation reaction between polyaldehydes and hydrazine. The first structure of its kind, labeled Py-Azine COF, was prepared by Dalapati et al. with 1,3,6,8-tetrakis(4-formylphenyl)pyrene (TFPPy) as a building block, a planar C_2_ tetraldehyde known for the rhombic geometry it induces in the crystalline structure [21] (Figure 3). The fact that hydrazine is a much less voluminous molecule than the complementing TFPPy monomer originates a favorable stacking between 2D layers, resulting in a particularly rigid, crystalline structure. Among several other applications, the potential of azine bond fluorescence has been reported as a chemical sensor for detection of specific compounds in aqueous solution [21].

β-ketoenamine-type COFs are a particular case of this class of materials, in the sense that the growth and crystallization processes develop in two different steps. The first step consists of the typical C=N reversible reaction between an aldehyde and an amine. The second step, however, is irreversible, and consists of a keto enol tautomerization, which results in an enamine bond [11]. The tautomerization occurs when the aldehyde used as a building block bears a hydroxyl group in an adjacent position to the reacting formyl group (bonded to the β-carbon). 1,3,5-triformylphloroglucinol (TFP) is an example of a regularly used monomer for this type of reaction.

In the tautomerization step, protons from the OH group migrate to the adjacent N atoms, resulting in an enamine group (Scheme 2). Since it is an irreversible reaction, β-ketoenamine COFs bear particular chemical resistance, keeping their structural integrity even in drastic pH conditions [1]. They are, however, typically less crystalline materials, due to the impossibility of conformational error correction once tautomerization occurs [11].

The first β-ketoenamine type COF, labeled TpPa-COF, was synthesized by using TFP and 4-phenylenediamine [22] (Scheme 2). Besides a regular, crystalline structure, the reported material also showed especially high chemical resistance, maintaining structural integrity in 9 mol·dm^−3^ HCl solution and 100 °C aqueous solution [22].

## 4. Substituting Groups and Post-Synthetic COF Modifications

Since the general COF applications imply the interaction of external compounds with the walls and internal surfaces of the pores and intrapore regions, it is paramount to determine and control crucial COF properties such as internal structure and intrapore chemical behavior.

The general structure and pore size is directly correlated to the dimensions and geometry of the building blocks [1]. The chemical environment depends on the intrinsic chemical properties of the surface atoms and the interactions they can establish with the guest molecules. To control the chemical properties and adapt them to specific, targeted applications, a viable solution consists of introducing substituting groups in the building blocks.

COF modifications can be introduced pre-synthesis, post-synthesis or in situ [1] (Scheme 3)**.** Pre-synthetic modifications consist of introducing adequate functional groups to the building block structures before the material is synthesized. A possible drawback of this approach is the fact that it is necessary to assure that the functional groups are stable in reacting conditions and do not interfere with the COF crystallization process, which may prove to be a difficult process depending on the functional groups and conditions.

In situ modifications mainly consist of using substrates that interact with the growing COF and directly interfere with the crystallization process itself [1]. By being present during the initial growth stages, these structures act as an anchor, allowing for the COF to polymerize around them, resulting in a composite material, which can be organic or inorganic, depending on the substrate.

One of the advantages of COFs, compared to other polymer classes, is that they allow for subsequent functionalization reactions even after they have been synthesized [1]. In non-porous polymer structures, post-synthesis reactions are difficult due to the poor solvent and reagent accessibility to the active reaction sites, as well as reduced mass transfer processes [12]. In the case of COFs, however, extended porous networks and high surface areas provide privileged access to solvent and reagents to the bulk of the solid, allowing for easy post-functionalization thus making this class of materials extremely versatile for a wide range of applications [1,11,12].

In order to achieve a covalent post-synthetic functionalization, the first step is to introduce reactive points in the starting monomers (Scheme 3). It is necessary to assure the COF has enough chemical stability in the new post-synthetic reaction conditions, which may eventually prove to be a difficult compromise (boroxine or boronic ester type COFs, for instance, are highly vulnerable to hydrolysis, making them very unstable in aqueous solvents).

Once the COF is prepared, the functionalization process consists of a heterogeneous reaction. The reaction should be favorable, high yielding and result in the least amount of secondary products [12]. Moreover, post-synthetic modification is a unique property of COFs that has earned this class the label of being simultaneously products and starting reagents for organic synthesis [12,16].

Another common strategy employed for post-synthetic functionalization consists of Cu(I) catalyzed click reactions [1,12]. Monomers bearing alkynyl groups in their structures are employed. Alkynyl groups do not react during the COF synthesis but are later activated for addition reactions. COF treatment with an organic azide in the presence of Cu(I) results in a cycloaddition reaction, with the formation of a heterocyclic 5-atom ring bearing the chosen functional group. *Click* reactions can usually be done under mild reacting conditions and are generally high yielding, and allow for the introduction of a wide range of functional groups, such as ethyl, acetate, hydroxyl, carboxyl or amine groups [1].

A different post-synthetic strategy for the introduction, for instance, of carboxylic groups, consists of ring opening reactions by using succinic anhydride [1,12]. COFs built from monomers containing OH functional groups are treated with succinic anhydride, which reacts with the hydroxyl groups, resulting in ester and carboxylic acid groups. The presence of acidic groups may eventually be detrimental to the regular COF crystallization process, which makes the post-synthetic functionalization of these groups even more relevant.

The introduction of amine groups in COF pore walls has also been explored for a wide range of applications. The presence of amine groups during COF synthesis usually interferes with COF formation and growth, namely imine or β-ketoenamine-type COFs, by distorting the projected symmetric structure. A viable solution consists of using monomers bearing nitro groups, and reducing them to NH_2_ groups, using metal catalysts, after the COF is prepared [1]. Treating the COF with SnCl_2_ allows for the easy formation of amine-functionalized COFs, a very useful process considering NH_2_ groups can potentially be the reacting sites for further functionalization.

Several other post-synthetic functionalization strategies for the introduction of different groups such as ether, thiol, amide, amidoxime or quaternary amines are summarized in.

A different approach consists in introducing structural modifications in the COF walls to alter the chemical properties within the porous regions. These strategies are useful when COF syntheses with certain linkage types are not viable or favorable; the preparation of an analogous COF, followed by the chemical conversion of its linkages can overcome the problem. Imine-type COFs, for instance, can be converted to amide-type structures through post-synthetic oxidation processes using sodium chlorite and 2-methylbut-2-ene, in a dioxane/acetic acid mixture; to thiazole-type COFs through treatment with sulfur; to benzoxazole-type COFs, by reaction with 2,5-diaminehydroquinone in the presence of water, DMF and molecular oxygen [1,12,24].

## 5. COF Applications

COFs combine a number of physical and chemical properties that are unique among the various classes of porous organic structures: high crystallinity, high BET surface area, customizable pore sizes and shapes, high stability in the vast majority of solvents, and unusually high thermal and physical stability under harsh conditions [1,16]. For these reasons, they are seen as very promising for the development of the most varied scientific, technological and industrial applications. In fact, in just a little over a decade after the first reported COF materials, there is already a vast amount of new synthesized or modified COF structures available in the literature, concerning the most varied type of applications, namely, gas storage and separation, heterogeneous catalysis, chemical sensing, controlled drug release, selective compound extraction for analytical quantification, ionic-exchange processes, photoconductivity, energy storage, electron carrying applications, proton conductivity or adsorption of pesticides and contaminants for residual water treatment [1,11,16]. In the following section, some of those applications will be highlighted, focusing on the structure-activity relationship.

### 5.1. Gas Adsorption

Gas storage was one of the first potential applications projected for COFs, largely due to the low density observed for these materials, owing to the high porosity and the fact that they are composed mainly by light elements, which allow for a superior gravimetric adsorption [1,16]. In the past decade, much research has been undertaken in order to implement COFs as gas storage materials for environmentally and technologically relevant compounds such as hydrogen, carbon dioxide, methane or ammonia.

Hydrogen is a clean energy source; however, its high instability and reactivity, even under controlled conditions, makes hydrogen, as it stands, a non-viable energy alternative for large scale distribution. COFs rise as one of the most promising class of materials for H_2_ storage [25].

Some of the COFs first reported were tested in hydrogen adsorption, among other gas molecules, and the influence of the pore structure and crystalline framework on the adsorption capacity was investigated [26,27,28]. It was observed that 3D COFs, which showed higher surface area and free volume compared to their 2D counterparts of similar structures, were generally better adsorbents, a fact that had been previously predicted through computational simulation studies [29]. In the case of COF-102, a 3D structure prepared by the auto-condensation of tetra(4-dihydroxyborylphenyl)methane (TBPM) (Scheme 4), an H_2_ maximum adsorption capacity of 77 mg g^–1^ was reported, at 35 bar and 77 K, a result rivalling the best MOFs and other porous structures obtained at similar high pressures, and almost meeting the current benchmarks established for implementing and building viable H_2_ fuel cells [26,27].

Monte Carlo and DFT simulations have been employed, at different levels of theory, to predict the interactions between H_2_ molecules and COFs constructed from different linkages and bearing pores from various sizes [30,31], as well as both 2D [32] and 3D COFs [33,34]. It has been concluded that, for the general test temperatures and pressures, the biggest contributing factor for adsorption efficiency is the available surface area and pore size, while specific H_2_-COF interactions and respective energies and heat of adsorption played a less significant role [30]. Among the works cited, theoretical H_2_ uptake values of up to 386 mg g^−1^ (100 bar, 77 K) [33] and 58.1 mg g^−1^ (1 bar, 298 K) [34] have been reported, the latter virtually reaching the established goal of 60 mg g^−1^ (6 wt%) uptake for practical room temperature applications, highlighting the enormous potential of these materials in the future.

Simulations for the adsorption of COFs impregnated with metal ions, mainly lithium [35,36,37], sodium, potassium [38] and transition metals [39] have also been carried out. It was observed that the interactions between H_2_ molecules and metal ions trapped within COF pores are more favorable and lead to adsorption capacities several times higher than those reported for metal-free COFs, at similar temperature and pressure conditions. Hydrogen uptake of 68.4 mg g^−1^ and 70.1 mg g^–1^ were reported, respectively, for Li-doped COF-105 (3D structure prepared from TBPM and HHTP) [37] and a COF containing C_6_ and B_2_C_4_ rings doped with scandium and titanium metal ions [39].

Likewise, several models were conducted for the adsorption of COF-320 bearing various substituting groups along the aromatic structure, such as hydroxyl, chloride, methyl or amine groups [40]. The latter group allows for significantly stronger interactions with the gas molecules, resulting in a 35% higher uptake compared to the non-functionalized COF. It could thus be concluded that introducing strategic substituting groups is an interesting approach to increase COF adsorption performance towards hydrogen and other gas molecules [40].

The unregulated accumulation of CO_2_ in the atmosphere due to urban and industrial emissions is one of the biggest current environmental problems, and one of the major causes of environmental phenomena such as excessive ocean acidification or greenhouse effect and consequent global warming. It is of utmost importance, therefore, to find practical, cheap materials for the selective capture and removal of CO_2_ from the atmosphere [41,42,43]. Early on, it was established that there is a direct correlation between the internal accessible surface area of COF structures and their efficiency in gas molecule trapping, including carbon dioxide [26,43]. In their pioneering studies, Furukawa et al. measured the adsorption capacity of several boronic ester and boroxine-type COFs, resulting in CO_2_ uptakes of up to 1180 mg g^−1^ for COF-102 (boroxine linkage, BET surface area of 3620 m^2^ g^−1^), at 55 bar and 298 K, surpassing several commercially used materials [26]. The promising results encountered for boronic acid COF derivatives led to investigating more chemically stable structures such as imine, azine, triazine or β-ketoenamine. Among the various examples found in the literature [41,42,43,44,45,46,47], Rabbani and coworkers reported a CO_2_ adsorption capacity of 1289 mg g^−1^ (40 bar, 298 K) for ILCOF-1 (Figure 4), a 2D structure synthesized from TFPPy and 1,4-phenylenediamine, showing a BET surface area of 2723 m^2^ g^−1^ [28].

It has been established that CO_2_ holds favorable interactions with nitrogen-containing molecules and structures, such as amine groups, which is why polyamine-based materials are currently used for CO_2_ trapping systems at industrial levels [43]. Thus, building nitrogen-rich COFs may be an interesting approach to develop for this application.

El-Mahdy et al. investigated the CO_2_ adsorption capacities for several imine-type 2D COFs synthesized from trialdehydes and triamines containing nitrogen atoms in heterocyclic aromatic rings [48] (Figure 5). The results showed that the planarity of the monomers have strict implication, not only on crystallinity, but also on the internal pore surface area. Regarding carbon dioxide tests, it was observed that the presence of nitrogen-containing groups significantly increased the COF adsorption capacity. Among the structures tested for comparison, the greater gravimetric uptakes were obtained for COFs containing the most nucleophilic nitrogen atom structures, in the most accessible positions within the pore, and with the least steric hindrance at the surface [48].

Further experiments with COFs prepared from monomers containing different structures were conducted. An and coworkers reported the synthesis and CO_2_ uptake of several 2D COFs resulting from the reaction between trialdehydes and diamines bearing carbazole and benzobisthiazole groups [49] (Figure 5). The structures, labeled Cz-COF and Tz-COF respectively, have S and N-rich heteroatom structures whose non-bonding electron pairs interact with guest carbon dioxide molecules, promoting their adsorption at the pore surface. At room temperature and pressure, uptakes of 110 mg g^–1^ and 154 mg g^–1^ were reported for Cz-COF and Tz-COF, respectively, as well as satisfactory recyclability and CO_2_ selectivity over N_2_, promising results compared to other classes of adsorbing materials [49].

Several other COFs containing heterocyclic groups such as triazine [50,51], carbazole [52,53] or thiadiazole [54], as well as post-synthetically modified COFs [55,56] are some of the various examples of the recently reported materials for CO_2_ capture and removal, proving the potential of these structures for the application.

COFs have also been investigated for the capture of other environmentally relevant gases such as ammonia—a toxic, corrosive compound and a health and environment hazard.

COF-10, a boronic ester linkage-type COF obtained by reacting HHTP and BPDA [57] (Figure 6) has been shown NH_3_ adsorption uptake up to 255 mg g^–1^ [57]. Boron atoms present within the structure act as Lewis acids, promoting favorable interactions with guest NH_3_ molecules.

A carboxylic acid-containing COF, labeled HOOC-COF, and Sr^2+^, Ca^2+^ and Mn^2+^ metal ions in its structure [58], also showed a high efficiency in the adsorption of NH_3_ [58].

Furthermore, COFs have shown potential for the removal and capture of other gas molecules, as methane and sulfur dioxide, among others [59,60,61,62,63,64,65,66]. One of these promising materials, Meso-COF-3, synthesized from the triamine with terephthalaldehyde, was one of the largest COFs regarding available pore parameters (4 nm pore diameter, 0.84 cm^3^ g^–1^ pore volume and a BET surface area of 986 m^2^ g^−1^) and showed an outstanding iodine adsorption capacity of up to 4000 mg g^−1^, at 75 °C [65].

### 5.2. Adsorption in Aqueous Solution

Excessive pollution due to human activity and agricultural and industrial production is one of the biggest environmental problems to face currently, and an enormous challenge for water and soil preservation [67]. Amongst the pollutants generally present in contaminated waters, some new emerging classes of organic compounds such as VOCs (volatile organic compounds) or PFASs (polyfluoroalkyl substances) have become gradually more concerning. Other organic pollutants such as food additives, biotoxins, dyes, pesticides and organic derivatives of cosmetic, hygiene or pharmaceutical products are also consistently found in contaminated waters and soils, as well as inorganic substances like heavy metals (chromium, nickel and mercury being some of the most concerning) or non-metals such as halides, cyanides, nitrates or even radionuclides [24]. The vast majority of these compounds have disastrous effects on human health, biodiversity and ecosystems, even at low concentrations; it is therefore crucial to find efficient, simple solutions to safely remove them and properly treat them [24,67,68,69,70]. Currently, several different techniques and processes are employed for the removal and degradation of contaminants from residual waters, such as coagulation, flocculation, sedimentation, chlorination, advanced oxidation processes, photochemical and electrochemical degradation, reverse osmosis, adsorption or filtration [24]. COFs have been getting considerable attention for the adsorption and removal of pollutants from water due to their intrinsic properties of chemical stability; high, permanent porosity and building versatility and customizability. 

COF adsorption of model dyes has shown very encouraging results. Zhu and coworkers reported the synthesis of CS-COF-1, a 2D imide-type structure based on 1,3,5-tris(4-aminophenyl)triazine (TAPT) and PMDA (Figure 7). Adsorption tests of methylene blue (MB), an industrial cationic dye, resulted in an adsorption capacity of up to 1691 mg g^−1^, higher than any other known porous material reported to date [71]. The structure showed pores of approximately 3.3 nm in diameter, wide enough to accommodate MB adsorbate (roughly 1.34 × 0.5 × 0.42 nm in dimensions).

Zhang et al. tested the potential of PAF-111, a 3D structure formed by C-C linkage by reacting TBPM and 1,3,5-tribromobenzene by Suzuki-Miyaura coupling reaction, in the removal of Rhodamine B in aqueous solution (Figure 8). A maximum adsorption capacity of 1666 mg g^–1^ was reported; once again, one of the highest results yet reported for other adsorbing materials towards the pollutant [72]. Of all the structures synthesized in that study, PAF-111 showed the highest available BET surface area (857 m^2^ g^–1^), highlighting the importance of this factor in the adsorption process.

Several other studies have also been reported regarding the adsorption of different dyes, originating mainly from the paper, paint and clothing industries, namely, Congo red (CR), methyl orange (MO), orange G (OG) and indigo carmine (IC). Polycationic COFs were synthesized for the removal of anionic dyes, with excellent adsorption capacities and recyclability [73,74], as well as COFs functionalized with electron-withdrawing groups for the adsorption of cationic dyes [75] and COFs impregnated with metal ions to promote favorable metal-dye coordination interactions [76].

Metal ion adsorption has also been a recent target of investigation for COFs, especially toxic metals released by heavy industry and other polluting sectors. Several experiments regarding Hg(II) adsorption have been reported, among other strategies [77,78,79], for COFs modified with S-rich groups, such as thiol [80,81,82] or thiomethyl [83]. TPB-DMTP-COF-SH, prepared from 1,3,5-tris(4-aminophenyl)benzene (TAPB) and a controlled mixture of 2,5-dimethoxyterephthalaldehyde, BDA(OMe)_2_, and 2,5-bis(prop-2-yn-1-yloxy)terephthalaldehyde, BDA(CH_2_C≡CH)_2_, has shown outstanding results. The BDA(CH_2_C≡CH)_2_ monomer contained C≡C groups that were post-synthetically reacted with 1,2-bis(2-azidoethyl)disulfane in order to create an S-rich functional group in the porous structure (Figure 9). In the presence of Hg(II) solution, TPB-DMTP-COF-SH showed maximum adsorption capacity of up to 4350 mg g^−1^, surpassing all other adsorbing materials reported to date [81].

Adsorption capacities of other heavy metals have also been reported, namely ions frequently found in industrial wastewater such as Cd(II) [84,85,86], Cr(VI) [87,88,89], Cu(II) [90,91] or Pb(II) [92,93], as well as Au(III) [94], lanthanides [95] and others [96,97,98,99,100,101,102,103]. 

The synthesis of a COF bearing EDTA groups for highly specific metal adsorption [96] (Figure 9) has been recently reported. The 2D β-ketoenamine structure, labeled TpPaNH_2_@EDTA, was synthesized via condensation reaction between TFP trialdehydes and 2-nitro-1,4-phenylenediamine, a diamine bearing nitro groups. The final COF material was obtained by two consecutive post-synthetic modifications: First, the nitro groups were reduced to NH_2_ groups by treatment in Na_2_S_2_O_4_ solution; then, the resulting amine groups were further modified via reaction with EDTA dianhydride. It was first observed that both the morphology and structure of the starting material remained intact post-modification. Moreover, due to the very strong, highly directed chelation effect of EDTA, good adsorption capacities were achieved towards several heavy metal ions, including Fe^3+^, Ni^3+^, Cu^2+^ and Pd^2+^, with up to 85% removal efficiencies within the first 5 min of sorption [96].

Organic compounds originating from agricultural activity, such as insecticides and pesticides, or from other specialized industries such as cosmetics and hygiene and cleaning products, are another great threat to water preservation. Among them, halogenated compounds are especially problematic and dangerous for the environment and human health. As such, COFs have recently been tested for the adsorption of polyfluorinated [104], polychlorinated [105] and polybrominated [106] compounds, showing good removal, selectivity and specificity properties. Ji et al. prepared an amine-functionalized COF to efficiently remove GenX, a perfluorinated pollutant [104]. The imine-linked 2D structure was synthesized by condensation between TAPB triamine and a dialdehyde containing azide-functionalized ethylene glycol. The azide groups were converted to NH_2_ groups through post-synthetic modification, yielding the final COF material (Figure 10). For a 20% degree of functionalization, a maximum adsorption capacity of 240 mg g^–1^ towards GenX was achieved, and at 28% functionalization, removal efficiency higher than 90% has been reached for several other perfluorinated compounds, confirming the positive influence of rationally implemented functionalization for specific molecule targeting.

Pharmaceuticals and drugs also constitute an ever increasing problem regarding water and environment preservation. Recently, Mellah and coworkers have assessed the adsorption performance of a new β-ketoenamine type COF (Figure 11) towards removal of common pharmaceuticals such as ibuprofen, diclofenac, acetaminophen and ampicillin [107].

The material, termed TpBD(CF_3_)_2_-COF, was synthesized by reacting TFP with 3,3’-bis(trifluoromethyl)-[1,1’-biphenyl]-4,4’-diamine, a fluorinated derivative of benzidine. The electron-withdrawing CF_3_ groups introduced in the structure showed particular affinity towards ibuprofen molecules, with a maximum adsorption capacity of 119 mg g^–1^ at neutral pH [107]. At pH = 2, adsorption capacity increases, since the drug is present in its protonated form, which promotes interactions with the fluorinated COF surface.

Tests of COFs bearing similar structures have also been reported for removal of other pharmaceuticals, mainly NSAIDs [108,109,110,111,112,113]. 

The structure named COF-SO_3_H (Figure 12) was obtained by the initial reaction between TAPB and 2,5-divinylterephthalaldehyde, followed by post-synthetic modification of the available vinyl groups with SO_3_H groups, by reacting the polymer with AIBN and ethane-1,2-disulfonic acid. The final material showed a maximum adsorption capacity of 770 mg g^−1^ towards diclofenac, higher than most other adsorbents reported thus far, including commercially available solutions, confirming the affinity of the drug towards sulfonate groups [110].

A vast amount of other organic pollutants have recently been tested, such as sulfamerazine, antibacterial agent and cosmetic pollutant [114]; furfural, a biomass and agriculture subproduct [115]; marine biotoxins [116,117]; flame retardants [118,119]; insecticides [120] as well as a wide range of pesticides [121,122,123].

Furthermore, several strategies to prepare composite materials between COFs and other ordered structures have recently been pursued. Composite COF membranes combine the permeability and structural resilience of the substrate with the inherent microporosity and high selectivity of COF layers, resulting in robust materials that can be used efficiently for a wide range of separation and filtration applications [124]. Composite COF membranes have successfully been synthesized using organic polymer structures such as polyacrylonitrile [125], polydopamine [126], polyamides [127], poly(1,1-difluoroethene) [128], poly(ether sulfone) [129,130], cellulose nanofibers [131], graphene [132], or even inorganic matrices such as alumina [133], with rather promising results in various adsorption and removal applications. Shen and coworkers devised a composite COF membrane for molecular separation in aqueous solution [126]. They used a polymeric polyacrylonitrile substrate previously treated with polydopamine (PDA). The deposited PDA layer behaved as a linker for subsequent COF nucleation. In the experiment, TFP and 2,5-diaminobenzenesulfonic acid were used as building blocks, resulting in a PDA-modulated crystalline ultrathin COF-PAN composite membrane (Scheme 5). Filtration measurements showed excellent water permeance of up to 1346 L m^−2^ h^–1^ MPa^–1^, as well as dye rejections of over 99% for several pollutants [126].

Furthermore, COFs have also been the subject for various other applications in analytical chemistry [134], most importantly, in solid-phase extraction processes [135,136,137,138,139].

### 5.3. Heterogeneous Catalysis

In heterogeneous catalysis processes, the catalyst and reagents are present in different physical phases, and the catalytic reaction occurs on the interface [140]. The catalyst (generally a solid) bears interfacial regions, called “active sites” that contribute to the activation of the catalytic reaction. A wide range of materials has been tested for the most varied type of reactions and syntheses [141]. COFs have recently earned the interest of researchers in this area: their high porosity, chemical stability and design versatility makes them ideal candidates for new and efficient solutions in heterogeneous catalysis [141,142].

One of the first studies for this type of application [18] was based on an imine-type polymer (Figure 13), further impregnated with Pd (II) ions post-synthesis. The obtained material has been used as a catalyst for Suzuki-Miyaura-type reactions. Conversion rates of up to 98% and very high rates of recyclability paved the way for further investigation in this area.

Due to their great utility in chemical synthesis, C-C coupling reactions have been vastly studied with a wide range of heterogeneous materials as catalysts, including COF structures. For example, Suzuki-Miyaura reactions carried out with porphyrin [143], triazine- [144,145] and triamine- [146,147,148] based COF structures as catalysts have been reported with promising results. All materials were rich in nitrogen atoms and were impregnated with Pd (II) post-synthetically prior to testing. The compound prepared by Yang and coworkers, COF-NHC, was a 2D imine-linked structure obtained via ionothermal reaction from 1,3,5-triaminobenzene and a dialdehyde modified with *N*-heterocyclic carbene groups as side chains, allowing for a highly specific coordination with palladium ions within the hexagonal pore walls (Scheme 6). High recyclability rates and conversions of up to 99% were obtained for Suzuki-Miyaura reactions at room temperature [146]. Besides Suzuki-Miyaura reactions, other C-C catalyzed coupling reactions have been reported, such as Heck [145,149,150,151,152] and Sonogashira [150,151], among others [153].

Other types of organic reactions have also been investigated. Salen-based COFs [154], as well as COFs modified with anionic groups such as sulfonic acid [155] or dodeca molybdophosphoric acid [156] have been used as catalysts for epoxidation of olefins, a useful strategy for obtaining synthetic precursors. The first material, coined COF-salen, is an imine-linked structure obtained from 1,3,5-tris(3′-tert-butyl-4′-hydroxy-5′-formylphenyl)benzene (TBHFPB) and ethylenediamine, bearing coordination points similar to those of salen ligands, allowing for extremely efficient, favorable intrapore metal ion coordination (Figure 14). In this type of reaction, the efficient coordination of the metal catalyst with the substrate is crucial for yield efficiency. The integration of Mg^2+^ and Co^3+^ metal ions, for instance, led to promising results for the catalytic epoxidation of styrene [154].

Carbon fixation is another type of reaction that has been gaining considerable interest within organic chemistry research. They consist of using CO_2_ gas, such as the collected emissions of industrial subproducts, as a reagent for the preparation of new, added value products, reducing the amount of CO_2_ released into the atmosphere and contributing to greener, more sustainable industrial processes. COFs have been tested in carbon fixation reactions with promising results [43]. Namely, CO_2_ cycloaddition to epoxides has been successfully reported for COFs containing hydroxyl [157,158,159], hydroxylamine [160] or ionic liquid derivatives [161,162,163,164] as substituting groups, as well as COFs impregnated with silver [165] or zinc [166] metal nanoparticles.

Other carbon fixation reactions using CO_2_ have been carried out [167,168] with COFs as catalysts, as well as several other types of industrially relevant organic reactions, such as cyanation [169], dehalogenation [170], dehydration [171], hydrolysis [172], condensation [173], esterification [174], oxidation [175,176], reduction [177,178,179,180], among others [181,182,183].

Asymmetric catalysis is another broadly studied branch in organic synthesis. Developing catalysts that are capable of transforming prochiral substrates into the desired enantiomer, with satisfactory enantiomeric excess and efficiency, is thus of great interest in organic synthesis, as well as industrial sectors that take the most advantage of chiral properties of molecules, such as cosmetics, care products, perfume or pharmaceutical industries [184].

As discussed above, COFs are generally modified post-synthetically with chiral groups for asymmetric catalysis applications [185]. Due to their specific, spatially defined orientation, those groups induce specific interactions with the reagents and substrates at the pore interface regions, which promote the specific formation of one enantiomer. Thus, a chiral COF for asymmetric catalysis applications has been synthesized [186]. The structure, named [(*S*)-Py]_x_-TPB-DMTP-COF, is a 2D imine-type framework in which a fraction of one of the monomers possess alkynyl groups, which were further reacted with (*S*)-pyrrolidine chiral groups post-synthesis (Figure 15). An important aspect to take into account while designing the COF synthesis is the density and distribution of functional chiral groups, in order to assure both the typical COF properties (such as porosity, stability and crystallinity) and catalytic efficiency of the new introduced groups. The prepared material was tested as catalyst for Michael addition reactions between cyclohexanone and nitrostyrene derivatives, where excellent conversion rates, enantiomeric excesses and recyclability have been reported [186].

Recently, several other COFs have been used for various applications in asymmetric catalysis. For instance, COFs containing pyrrolidine derivatives for Steglich rearrangement [187] and aldol condensation [188]; TADDOL ligand derivatives for aldehyde alkylation with ZnEt_2_ as metal catalyst and alkylating agent [189]; and salen derivatives for cyanation of aldehydes, Diels-Alder reactions and epoxidation of olefins [190].

COFs have also been used for electrocatalysis and photocatalysis, both of which are generally regarded as sustainable, greener alternative solutions of lower energy costs. The intrinsic mass transfer and charge carrying properties of most COFs, due to the highly reticulated, mesoporous frameworks and conjugated bonds, are a great advantage for applications in those areas.

For electrocatalysis, for instance, COFs showing good mass transfer properties were reported for the electrocatalytic production [191] and reduction [192] of molecular oxygen.

Similarly, light absorption and charge transfer properties shown by aromatic COFs have recently been studied for applications in photocatalysis. Aromatic COFs usually have high absorptivity in the visible and UV spectra and show charge transfer processes consisting of electron delocalization within the conjugated structure, inducing electron-hole pair formation [193]. This phenomenon, typical of semiconducting materials, is due to the closeness between conduction and valence band energy levels, allowing for the formation of extremely reactive radical species at the surface of the material, thus initiating the photocatalytic process.

COF structures used in photocatalysis are generally obtained from aromatic building blocks, thus inducing aromaticity throughout the whole structure allowing for increased free radical formation. LZU-190, prepared by Wei and coworkers from TFB and 2,5-diaminohydroquinone, is an example of such a structure [194] (Figure 16). LZU-190 is a benzoxazole-type framework showing up to 99% yields for oxidative hydroxylation of arylboronic acids to the respective phenols, under visible light irradiation in the presence of molecular oxygen [194].

For economical and sustainability reasons, visible light-catalyzed reactions are of special relevance from a synthetic standpoint. Various other experiments have thus been reported for photocatalysis applications in the presence of COFs, such as oxidation of amines [195], alcohols [196] or sulfides [197]; as well as other types of reactions, such as C-C coupling reactions [198,199], isomerization of olefins [200], carbon dioxide [201] and heavy metal [202] reduction or pollutant degradation [203].

Several water splitting studies have also been reported using COFs as photocatalysts. The reaction consists of obtaining molecular hydrogen and oxygen from water, a process of great interest for numerous technological and industrial applications, since, theoretically, it can be one of the most efficient, practical methods for H_2_ production. COFs with different structures and linkage types, such as triazine [204,205,206,207,208], β-ketoenamine [209,210,211,212], hydrazine [213], imine [214,215] or azine [216] have been used. For instance, Biswal and coworkers reported the synthesis of TpDTz COF for photocatalytic hydrogen evolution reactions [210]. The β-ketoenamine-linked 2D structure was synthesized from TFP and 4,4′-(thiazole [5,4-*d*]thiazole-2,5-diyl)dianiline, a C_2_ symmetric diamine bearing thiazole groups in its backbone (Figure 17). The unusually high, light-absorbing capabilities of the material allowed for long-term, stable hydrogen production with no efficiency loss, at a maximum rate of 941 µmol h^−1^ g^−1^, as well as satisfactory turnover number (TON), rivalling and surpassing many previously reported water reduction systems [210].

### 5.4. Chemical Sensors

Much like the research developed with respect to catalysis, the potential of COFs for applications related to chemical sensing has recently been broadly investigated. Photoluminescence properties intrinsic to aromatic COFs, due to the conjugated double bonds and π electron delocalization, as well as the rigid structure usually obtained by the use of planar, aromatic monomers have raised the interest in this class of materials for chemical detection and related applications. Complexation of COFs with certain guest species can lead to the fluorescence quenching, or to the activation of fluorescence channels otherwise unavailable in the solid in its free form. This means that COFs can be rationally designed and molded specifically for selective photodetection of compounds [217]. An example of this is the synthesis of a fluorescent COF for the chemical detection of pollutants in aqueous solution [218]. The imine-linked structure, named IMDEA-COF-1, was obtained via condensation reaction between TFB and 1,6-pyrenediamine (PyDA) and showed a crystalline, hexagonal framework (Figure 18). The complexation of dissolved dyestuff, such as MB, inside COF pore walls resulted in reduced fluorescence intensity, which allowed for the selective detection of these types of compounds.

Besides dyestuff, COFs have also been developed for the selective detection of metals [219,220], a wide range of organic compounds [221,222,223,224,225,226,227,228,229,230], biomolecules [231,232,233,234,235,236,237], gases [238,239,240], water and humidity [241,242,243] or even pH detection [244,245]. For instance, a COF for highly selective detection of explosives [222] has been developed. The imine-linked structure, Py-TPE-COF, was synthesized from TFPPy and non-planar 1,1,2,2-tetrakis(4-formylphenyl)ethane and exhibits permanent pores 1.1 nm wide (pore volume 0.77 cm^3^ g^−1^) and a BET surface area of 987 m^2^ g^−1^. The highly photoactive COF showed over 21% absolute photoluminescence quantum yields and selective luminescence quenching in the presence of 2,4,6-nitrophenol (TNP) (Figure 19), proving the potential of COFs for fluorescence-sensing applications [222].

Another interesting example are hydrazone-linked COFs for photosensitive detection of Fe^3+^ in aqueous solution [220]. The structures bear strategic oxygen-nitrogen-oxygen chelating sites that favor intrapore metal coordination, namely with Fe^3+^, with outstanding selectivity and sensitivity. In a similar approach, Cui and coworkers interestingly prepared COF nanosheets for Al^3+^ ion detection [219]. The as-synthesized Bpy-COF, prepared from TFP and [2,2′-bipyridine]-5,5′-diamine, initially shows no luminescent behavior; however, upon exfoliation via grinding and ultrasonic techniques, the resulting COF nanosheets become moderately responsive to UV-light radiation. It was observed that the material was highly sensitive to the presence of Al^3+^ in solution, with the fluorescent intensity sharply increasing as Al^3+^ concentration increases, unlike with several other alkaline or transition metals tested, due to overall weaker coordination interactions with bipyridine groups.

Peng et al. successfully synthesized a highly crystalline COF for sensitive and selective DNA detection [233]. The 2D imine-linked structure, prepared by condensing TPA triamine with tris(4-formylphenyl)amine, showed a BET surface area of 1136.5 m^2^ g^−1^, 0.89 cm^3^ g^−1^ pore volume and average pore size of 1.5 nm. The bulk solid was then processed into ultrathin nanosheets via exfoliation by solvent dispersion and ultrasonic techniques, prior to biosensor experiments. Designed DNA probes marked with a fluorescent dye adhere to the pore wall surface which results in the quenching of the COF’s fluorescence. However, when target DNA is present in solution, a chain reaction between the probes and the DNA strand is triggered, and the resulting *ds*DNA biomolecule, which has very weak bonding interactions with the COF, leaves its surface, thus “switching on” the COF’s fluorescent activity again. This predictable, linear behavior allows for a quantitative detection of DNA and can potentially be used for many other biosensing applications [233].

### 5.5. Electronic Applications

The promising results obtained for COFs regarding electrochemical energy storage and ionic conductivity also turns these materials into potential candidates in a wide range of new technological applications such as fuel cells or solid-state ionic batteries [246]. In order to explore these properties for applications in proton conductivity, several aromatic COFs have been synthesized and further impregnated with acidic compounds, such as phosphoric acid or phosphoric and sulfonic acid derivatives [247,248,249,250,251]. Banerjee et al. successfully devised a self-standing COF membrane with high proton conductivity [249]. A 2D β-ketoenamine-linked structure was obtained via condensation between TFP and 3,3′-dimethylbenzidine. The reaction was mediated by *p*-toluene sulfonic acid monohydrate, which acted as a co-reagent allowing for a slower crystallization process, thereby inducing better porosity and crystallinity properties. The entrapped acid compound also acted as an efficient proton carrier for conductivity testing. The resulting COF membrane, obtained by a slow reaction process at low temperatures for a long period (3 to 4 days) to allow for thermodynamic control and consistent framework error correction, showed high crystallinity and an extremely high proton conductivity of up to 7.8 × 10^−2^ S cm^−1^ at 80 °C and 95% relative humidity, higher than the majority of organic polymer materials reported to date [249]. In a different approach, 2D COF structures bearing proton-donating groups in the porous framework were designed [252]. One of the reported materials, NUS-10, prepared via a liquid-assisted mechanochemical grinding procedure between TFP and 2,5-diaminobenzene-1,4-disulfonic acid, showed a 69 m^2^ g^–1^ BET surface area and 0.84 nm pore size. In its free form, a proton conductivity of up to 3.96 × 10^–2^ S cm^–1^ was reported for NUS-10, at 298 K and 98% relative humidity, one of the highest reported values for similar structures. Furthermore, a mixed-matrix COF membrane consisting of a blend of the resulting COF and nonconductive polyvinylidene fluoride (PVDF) polymer was prepared, affording up to 1.58 × 10^−2^ S cm^−1^, in water at 353 K [252].

Several COFs have been further tested for implementation as novel lithium [253] and lithium-sulfur composite batteries [254,255,256], as well as cheaper, more sustainable alternatives to lithium, such as sodium- [257] and potassium- [258] based batteries. COFs have also drawn the attention of researchers for application as organic photovoltaic materials. Ideal solar cell substrates combine high absorbance for maximum light absorption with fast charge diffusion, transfer and separation at the framework surface; thus COFs, owing to their stability, absorption and interesting semiconducting and charge carrying properties are potential candidates for implementation as photovoltaic cells [259]. Wu et al. developed 3D imine-linked COFs for perovskite solar cell enhancement [260]. One of the materials, SP-3D-COF-2, was obtained from 3,3′,6,6′-tetramine-9,9′-spirobifluorene and 4,4′-biphenyldicarbaldehyde. The tetramine gives the resulting material a two-dimensional unit cell of rhombic topology (Figure 20); however, the experimentally obtained solid showed a 3D framework resulting from 2D layer interpenetration. A high BET surface area of 1582 m^2^ g^−1^ and pore size and volume of 1.5 nm and 0.97 cm^3^ g^−1^ were reported, respectively. The optically-active COF was further tested by using it as a dopant for a reference perovskite solar cell, substantially improving power conversion efficiency by up to 18.0% as compared to the non-doped material [260].

Moreover, besides photovoltaic cells [261,262], several other reports of COFs being tested for electronic and electrochemical energy storage-related solutions have recently appeared [263,264,265,266,267,268,269], once more proving how promising this new class of materials can potentially be for future practical applications.

### 5.6. Biomedical Applications

Researchers have recently discovered the potential of COFs for use in biomedical applications, such as drug delivery and cancer therapy [270,271,272]. The high porosity, pore volume and surface area typically observed in synthesized COF structures allow for high drug loading capacity, making them good candidates as drug delivery agents. Moreover, versatile pore wall tunability via pre or post-synthetic modifications allows for the preparation of customized COFs that are capable of efficient drug release with a high degree of control over diffusion and release rates. Additionally, the high covalent reversibility found in many COF structures can be used as an advantage for biomedical applications, in the sense that COFs can be specifically designed, through rational monomer and linkage selection, to strategically biodegrade after completing its task inside the organism. Recently, Oliveira et al. prepared a novel amino-functionalized COF-5-based structure for the controlled release of antitumor drug camptothecin [273]. A novel one-pot synthetic approach was employed in which NH_2_-functionalized boronic acid monomer (BDBA) replaced the non-functionalized counterpart (BDBA) via sequential monomer substitution. The amine functionalization allowed for the conjugation of camptothecin-20-*O*-hemisuccinate (CPT-Suc) within the COF pore walls. Characterization showed that the CPT ligand increased hydrophobicity of the material, therefore increasing COF stability in water—an important aspect to consider as boronate ester COFs are particularly susceptible to hydrolysis. More importantly, the CPT-COF conjugate was observed to be easily internalized by cell membranes and showed higher cytotoxic activity than the free drug, even at low doses [273], adding another example to a vastly growing amount of reports of COF materials as promising drug delivery agents [271,274,275,276,277]. Guan et al. designed and prepared a functionalized COF nanocomposite for highly selective synergistic tumor therapy [278]. The final COF-based structure, termed CaCO_3_@COF-BODIPY-2I@GAG, was obtained by first reacting 2,5-dimethoxyterephthalaldehyde (DMTP) with 1,3,5-tris(4-aminophenyl)benzene (TAPB). The 2D imine-linked COF was then functionalized with the amine-substituted BODIPY-2I. The functionalized material was further modified through the sequential impregnation of CaCO_3_ and glycosaminoglycan (GAG), yielding the final composite. BODIPY-2I acted as photosensitizer in a photodynamic therapy experiment, whereas GAG served as the target agent for CD44 receptors on the digestive tract tumor cells. The synergistic effect of highly reactive singlet oxygen (^1^O_2_), produced by the photosensitive BODIPY-2I, combined with Ca^2+^ overload promoted by the loaded COF leads to tumor cell destruction. In vitro experiments showed selective tumor cell death induced by both Ca^2+^ overload and ^1^O_2_ oxidative stress by using green light as light source, suggesting the efficiency of COF nanomaterials for the novel approach [278]. Several other studies have been recently reported, attesting the viability of COFs as active cancer therapy agents [279,280,281,282,283,284].

## 6. Conclusions

It is undeniable that COFs are a very promising new class of porous materials with enormous potential for a wide range of applications, already showing impressive results in several different fields of study and research that can rival and even sometimes surpass already established, well-known, commercially available solutions. However, COF development is still in its very early stages and a few serious challenges still need to be addressed and overcome before COFs can be considered a viable alternative for mass-scale real-use applications. First and foremost, the “crystallization problem” needs to be tackled and properly solved. COF synthesis needs to be standardized, consistent and reproducible. This means researchers need to acquire more knowledge and information regarding the specific mechanisms of COF formation, growth and crystallization, in order to more precisely control the polymerization processes and more accurately take advantage of desired physical properties such as porosity, surface area and crystallinity. Then, for the scale-up challenge, if COF materials are to be employed in the industry on a massive scale, standardized synthetic procedures must first be achieved. This implies discovering new synthetic pathways that allow for a consistent COF material with intact properties in easier, more efficient, faster and cheaper ways. Despite these challenges, however, COFs are set to remain a very encouraging, exciting prospect in materials chemistry.

## Data Availability

No new data were created or analyzed in this study. Data sharing is not applicable to this article.

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
