# Peer review of "Covalent Organic Frameworks: Synthesis, Properties and Applications—An Overview"

_polymers, 2021, doi:10.3390/polym13060970_

Round 1

Reviewer 1 Report

This review manuscript by Valente et al. covers a wide range of information related to COFs including synthesis, property, applications. The author has done a great job for the purpose. The manuscript could be handy for scientists who are currently players in the related research and are interested in knowing more about the subjects.

Author Response

We are really grateful for such positive comment.

Reviewer 2 Report

The authors review the synthesis and applications of Covalent Organic frameworks, an interesting topic in materials chemistry. Many reviews about these materials have been reported in the last years covering different aspect of these networks. The review is correct, however I think it should cover new issues related to these frameworks, so the authors should improve the text.

For instance, the review does not cover new linkages used in the synthesis of COFs and I think they should be indicated. The same for other applications that at least they should be mentioned as the use, for instance in cancer therapy, etc…

Otherwise, frameworks named as COTs (Scheme 7) should not be included as COFs. Properly, Covalent Organic Frameworks (COFs) must be crystalline to be considered as real COFs as Professor Yaghi claims.

Reviewer 3 Report

This manuscript presents a comprehensive overview related to the synthesis, properties, applications of emerging covalent organic frameworks. Since COFs combine high, permanent porosity and surface area with high thermal and chemical stability, crystallinity and customizability, they are becoming ideal candidates for a myriad of promising new solutions in a vast number of scientific fields, with widely varying applications such as gas adsorption and storage, pollutant removal, degradation and separation, advanced filtration, heterogeneous catalysis, chemical sensing or energy storage and production and a vast array of optoelectronic applications. This review can bring us a systematical understanding of COFs from their history, design principle, and modification strategy. The topic is very interesting and the whole manuscript is prepared quiet well. Therefore, it deserves publication after minor revision. The specific comments can be found as below. (1) Some content discussed in different sectors is overlap. For example, section 2 and 3 both talk about the synthesis of COFs. The authors should clarify the key point of each part because the overlap may confuse the readers. (2) Although this manuscript is an overview, the language should be more concise and clearer. It should be well presented rather than a chapter of a book. (3) References in section 1 regrading Reticular Chemistry and MOFs should be increased. Specially, as the first material using reticular chemistry, MOFs should be discussed more in terms of their fabrications, modulations, and applications. Here are some useful references: Chemical Engineering Journal, 2021, 404, 127075; Journal of Hazardous Materials, 2019, 367, 456-464.
